# Evaluating Ocular Response in the Retina and Optic Nerve Head after Single and Fractionated High-Energy Protons

**DOI:** 10.3390/life11080849

**Published:** 2021-08-19

**Authors:** Xiao-Wen Mao, Seta Stanbouly, Tamako Jones, Gregory Nelson

**Affiliations:** Department of Basic Sciences, Division of Biomedical Engineering Sciences (BMES), Loma Linda University School of Medicine and Medical Center, Loma Linda, CA 92350, USA; sstanbouly@llu.edu (S.S.); tkjones@llu.edu (T.J.); grnelson@llu.edu (G.N.)

**Keywords:** space radiation, protons, mouse retina, optic nerve head, blood-retinal barrier, fractionation dose, apoptosis, oxidative stress

## Abstract

There are serious concerns about possible late radiation damage to ocular tissue from prolonged space radiation exposure, and occupational and medical procedures. This study aimed to investigate the effects of whole-body high-energy proton exposure at a single dose on apoptosis, oxidative stress, and blood-retina barrier (BRB) integrity in the retina and optic nerve head (ONH) region and to compare these radiation-induced effects with those produced by fractionated dose. Six-month-old C57BL/6 male mice were either sham irradiated or received whole-body high energy proton irradiation at an acute single dose of 0.5 Gy or 12 equal dose fractions for a total dose of 0.5 Gy over twenty-five days. At four months following irradiation, mice were euthanized and ocular tissues were collected for histochemical analysis. Significant increases in the number of apoptotic cells were documented in the mouse retinas and ONHs that received proton radiation with a single or fractionated dose (*p* < 0.05). Immunochemical analysis revealed enhanced immunoreactivity for oxidative biomarker, 4-hydroxynonenal (4-HNE) in the retina and ONH following single or fractionated protons with more pronounced changes observed with a single dose of 0.5 Gy. BRB integrity was also evaluated with biomarkers of aquaporin-4 (AQP-4), a water channel protein, a tight junction (TJ) protein, Zonula occludens-1 (ZO-1), and an adhesion molecule, the platelet endothelial cell adhesion molecule-1 (PECAM-1). A significantly increased expression of AQP-4 was observed in the retina following a single dose exposure compared to controls. There was also a significant increase in the expression of PECAM-1 and a decrease in the expression of ZO-1 in the retina. These changes give a strong indication of disturbance to BRB integrity in the retina. Interestingly, there was very limited immunoreactivity of AQP-4 and ZO-1 seen in the ONH region, pointing to possible lack of BRB properties as previously reported. Our data demonstrated that exposure to proton radiation of 0.5 Gy induced oxidative stress-associated apoptosis in the retina and ONH, and changes in BRB integrity in the retina. Our study also revealed the differences in BRB biomarker distribution between these two regions. In response to radiation insults, the cellular response in the retina and ONH may be differentially regulated in acute or hyperfractionated dose schedules.

## 1. Introduction

Space radiation from galactic cosmic rays (GCR) and solar particle events (SPE) will pose significant health risks, especially for long-duration lunar and Mars missions. GCR are continuous low-dose-rate exposures composed of approximately 87% protons, 12% helium ions, and 1% high charge, high energy (HZE) particles of Z > 2 and will comprise the majority of space radiation exposure for long- duration missions beyond the Van Allen belts [1]. The space radiation environment also consists of mixtures of high-energy protons and a minor fraction of other ions released from the sun during SPEs. During long-term deep space missions, it is anticipated that multiple SPEs will be encountered. We and others have been studying the effects of space radiation on the ocular tissue in animal models using beams of high-energy protons produced by particle accelerators [2,3,4,5]. 

Prolonged spaceflight induces physiologic and pathologic neuro-ophthalmic abnormalities in astronauts, collectively defined as the Spaceflight Associated Neuro-ocular Syndrome (SANS) [6]. However, the underlying mechanisms of the changes and factors contributing to the development of damage are unclear. Studies have shown that radiation at doses less than 1 Gy induced significant levels of retinal oxidative damage including retinal cell apoptosis [4]. More recent reports showed optical nerve swelling in astronauts and pathological changes around the optical nerve disc including prelaminar region, lamina cribrosa, and retro-laminar region [7]. The mechanisms and factors that contribute to the development of optic disc edema are less known and not well investigated. Recognizing the importance of the health and performance of astronauts both during and post-flight, we believe that understanding the impact of space radiation on the retina and optic nerve head (ONH) is crucial in assessing the risk to astronauts during long-duration missions, especially for future deep-space missions that will expose astronauts to greater radiation than previously encountered [8]. However, the effects of spaceflight relevant dose and dose rate exposures to protons in the mouse retina and in the ONH have not been previously addressed. 

The goal of the present study was to investigate the effects of whole-body proton exposure at a single dose of 0.5 Gy on apoptosis, oxidative stress, and blood-retina barrier (BRB) integrity in the retinal and ONH region and to compare these radiation-induced effects with those produced by fractionated doses for a total dose of 0.5 Gy in 12 equal fractions over 25 days to simulate the chronic exposures in space. 

## 2. Material and Methods

### 2.1. Animals and Irradiation with High Energy Protons

Six-month-old C57BL/6 male mice, each weighing between 30–35 g, were purchased from the Jackson Laboratory (Bar Harbor, ME, USA) and shipped to Loma Linda University (LLU) in Loma Linda, CA. The animals were maintained under a constant temperature of 68 ^°^F with a 12-h day/night cycle. Chow and water were available ad libitum. After a one-week acclimation period, the mice were either sham irradiated or received whole-body high energy proton irradiation at a single dose of 0.5 Gy or 12 equal dose fractionations for the total dose of 0.5 Gy over 25 days. To simulate a space-like energy distribution of protons, we have used a spread-out Bragg peak set-up. To facilitate frequent use of medical treatment facilities, we selected energies and field modulations that are commonly used in cancer treatment at the J.M. Slater Proton Treatment Facility at Loma Linda University Medical Center, so that dosimetry, beam tuning, and other practical considerations would be relatively routine and reproducible. This facility uses a synchrotron built at Fermi National Laboratory upgraded and operated by Optivus Proton Therapy, Inc. as the Conforma 3000^®^ Proton Beam Treatment System. The operating and dosimetry system is Odessey™ provided by PerMedics, Inc., of San Bernardino, California, USA. Dosimetry is provided by a series of parallel plate ion chambers calibrated to a NIST-traceable thimble ion chamber [9].

Therefore we used a beam of tuned energy 186 MeV and passed the beam through ~12 cm of plastic (“solid water”™) to reduce energy to 140 MeV and a spinning modulator wheel to produce a field of constant dose over a depth of 14 cm resulting in a field of protons with energies from 0 to 140 MeV in approximately equal abundance with high spatial uniformity. The range of LET values in the field is from 0.57 at the entrance to ~21 keV/µm at component Bragg Peaks and samples occupied the first 7.5 cm of the spread-out Bragg peak.

Proton exposures were performed Monday, Wednesday, and Friday evenings after patient treatments. The acute exposure was a single exposure to 0.5 Gy while for the fractionated exposures there were 12 fractions each of 4.17 cGy. The acute exposures were 38–43 s long at a dose rate of 0.74 Gy/min (clinical dose rate) while the fractions were 30–37 s long delivered at 1/10th the clinical dose rate at 0.075 Gy/min in 13–16 accelerator spills. Acute exposures were performed only during the 12th irradiation session. For each exposure, there were two 10 × 10 × 7.5 cm ventilated plastic restraint boxes each containing two freely-moving animals from the same cage. The boxes were aligned to a 20 cm circular field target placed on the patient treatment table where the beam passed vertically downward through the target avoiding self-shielding by the animals. Animals were loaded into restraint boxes in the animal care facility and transported by covered cart to the proton treatment area. Small battery-powered fans operated inside the cart to assure adequate air circulation. Total time in restraint boxes was approximately one hour. To assure equal restraint and handling stress across the three treatment groups, sham irradiated mice were subjected to the same restraint and transportation procedures during each of the 12 irradiation sessions. Similarly, acute irradiated animals were restrained and transported during each irradiation session but only exposed to protons during the last session. Total time in restraint boxes for all animals was approximately one hour on each of the twelve procedure days. Restraint stress precautions were driven by post irradiation behavioral testing needs. All procedures were approved by the Radiation Safety and Institutional Animal Care and Use Committee at LLU.

### 2.2. Eye and Retina Preparation

At four months after proton exposure and periodic behavioral testing, mice were euthanized and the right eye from each mouse was placed individually in sterile cryovials, snap-frozen in liquid nitrogen, and kept at −80 °C prior to use. The left eyes were fixed in 4% paraformaldehyde in phosphate-buffered saline (PBS) for immunohistochemistry (IHC) assays. 

### 2.3. Terminal Deoxynucleotidyl Transferase Dutp Nick End Labeling (TUNEL) Assay

Sections were cut coronally through each left eye, and sections were roughly 100 μm apart providing five sections per eye for analysis. The ocular sections were evaluated using the TUNEL assay according to standard procedures. Briefly, six μm paraffin-embedded sections were processed using DeadEnd™ Fluorometric TUNEL system kit, Promega Corp., Madison, WI, USA) The same sections were then incubated with DyLight 594 Lycopersicon esculantum-Lectin (Vector Laboratories, Burlingame, CA, USA) to stain the endothelium. Nuclei were counterstained with diamidino-2-phenylindole (DAPI, blue, Life Technologies, Eugene, OR, USA). 

### 2.4. Immunostaining for 4-Hydroxynonenal (4-HNE)

Immunofluorescence staining for an oxidative damage marker on ocular sections was performed using 4-HNE antibody specific for lipid peroxidation. Six μm sections were incubated with the anti-4-HNE antibody (Alpha Diagnostic International Inc., San Antonio, TX, USA) at 4 °C for overnight followed by a rabbit anti-rabbit IgG fluorescence-conjugated secondary antibody ( Invitrogen Corp., Waltham, MA, USA) for 1.5 h at room temperature and counterstained with DAPI.

### 2.5. Immunohistochemistry for Aquaporin-4 (AQP-4) and Vascular Double-Labeling

Six μm sections were incubated overnight (18–21 h) at 4 °C with primary antibodies polyclonal rabbit anti-AQP-4 (Santa Cruz Biotechnology, Inc. Dallas, TX, USA) and monoclonal mouse anti-glial fibrillary acidic protein (GFAP) clone GA5 (Millipore, Burlington, MA, USA) in 0.25% BSA, 0.25% Triton X-100 in PBS. Sections were washed three times in PBS and further treated with secondary antibodies Alexa Fluor 488 goat anti-rabbit IgG and Alexa Fluor 568 Goat anti-mouse IgG (Life Technologies). Cell nuclei were counterstained with DAPI.

### 2.6. Immunostaining Assays for Platelet Endothelial Cell Adhesion Molecule (PECAM-1) and Zonula Occludens-1 (ZO-1)

Immunofluorescence staining related to BRB integrity on ocular sections was performed using biomarkers against PECAM-1 and ZO-1. Six µm sections were deparaffinized in Histoclear, rehydrated and, washed in PBS for 20 min. Vasculature was labeled with DyLight^®^ 488 *Lycopersicon Esculentum* (Tomato) Lectin (Vector Laboratories) for 30 min at room temperature followed by 10 min wash in PBS. Sections were then incubated overnight at 4 °C with primary rabbit antibodies against ZO-1 (Thermo Fisher Scientific, Hampton, NH, USA) or CD31/PECAM-1 (Novus Biologicals, Centennial, CO, USA) in antibody dilution buffer. After 3 washes in PBS, sections were incubated for 1.5 h with secondary antibody goat anti-rabbit IgG Alexa Fluor^®^ 568 (Life Technologies) followed by PBS washes. The cell nuclei were counterstained with DAPI. 

### 2.7. Quantification of Immunostaining

Six to ten field images were examined using a BZ-X710 All-in-One inverted fluorescence microscope with structural illumination (Keyence Corp., Elmwood Park, NJ, USA) at 20× magnification spanning the entire retina per section. Data were obtained for 6 ocular samples of each group. For quantitative analysis, numbers of TUNEL positive cells were counted within the entire retina layer and in the ONH. The total number of TUNEL-positive nuclear/ganglion cells or endothelial cells in the retina or ONH were counted in five sections for each eye. The surface of each section was measured on digital microphotographs using ImageJ v1.49v software (available as freeware from National Institutes of Health, Bethesda, MD, USA; http://rsbweb.nih.gov/ij/, access date: 1 June 2018) and the density profiles were expressed as a mean number of apoptotic cells/mm^2^. The mean of the density profile measurements across 5 retina sections per eye was used as a single experimental value.

To determine AQP4, ZO-1, and PECAM immunoreactivity, fluorescence intensity was measured on six to ten randomly selected fields on the retina or in the ONH of each section and calculated using ImageJ software. Fluorescence intensities for positive cells (red channel for ZO-1 and PECAM, and green channel for AQP-4) from the areas of interest were measured and data were extracted and averaged across five ocular sections per eye within the group as one experimental value.

### 2.8. Statistical Analysis

The results for the TUNEL assay and IHC were analyzed by one-way analysis of variance (ANOVA) and Tukey’s HSD (honestly significant difference) test for multiple pair-wise comparisons (Sigma Plot for Windows, version 13.0; Systat Software, Inc., Point Richmond, CA, USA). Means and standard error of means (SEM) are reported. α-level was set at 0.05 for all tests of statistical significance.

## 3. Results

### 3.1. Apoptotic Damage in the Retina

Quantitative assessment with TUNEL assay indicated that the density of apoptotic cells in the retinal nuclear layer and ganglion cell layer (GCL) was significantly increased (*p* < 0.05) in the irradiated groups by roughly 65% compared to that in the control group (Figure 1A). The number of TUNEL positive cells was also calculated in the ONH region (Figure 1B). A significant increase of apoptosis in retinal cells was detected in the fractionated group by 50–67% compared to radiation with a single dose and sham controls, respectively (*p* < 0.05). 

### 3.2. Oxidative Damage Biomarker Using 4-HNE

Immunofluorescence staining for an oxidative damage marker was performed using an antibody against 4-HNE specific for lipid peroxidation. A significant increase in immunoreactivity of HNE was seen in the retina (Figure 2A) of the mice that received a single dose compared to sham controls (*p* < 0.05). A strong trend increases (*p* = 0.06) in the fractionated dose group was also documented compared to controls (Figure 2B). In ONH (Figure 2C), the level of anti-HNE antibody protein in irradiated groups was significantly higher than controls. The immunoreactivity of HNE was more pronounced in the acute single dose group compared to the fractionated mouse group (Figure 2D). 

### 3.3. Alteration of BRB Integrity Using Water Channel Protein AQP4, Adhesion Molecule PECAM-1, and Tight Junction Protein ZO-1

AQP4 is a water channel protein concentrated at the luminal surfaces of astrocyte end-feet, which plays an important role in regulating of water permeability. No significant differences in protein expression of glial fibrillary acidic protein (GFAP), a biomarker for astrocyte activation, was noted between groups. However, increased AQP-4 staining was seen in the irradiated groups compared with controls in the retina (Figure 3A). The difference in AQP-4 expression was greater in the irradiated retina of an acute single dose (*p* < 0.05) than that of fractionated one (*p =* 0.07) (Figure 3B). Low levels of AQP-4 expression were detected in the ONH region for all groups with no significant difference between groups (data not shown). 

Tight junction proteins and cell adhesion molecules play important role in regulating BRB integrity. Protein expression of PECAM-1 and ZO-1 was evaluated in the retina and ONH region. In the control retinal tissue, only some positive cells for PECAM-1 were found. In the irradiated retina from both single and fractionated dose groups, enhanced immunoreactivity of PECAM-1 cells was apparent in the retinal inner nuclear layer (INL) and GCL (Figure 4A). Irradiated groups (single or fractionated doses) showed a significant increase of PECAM immunoreactivity compared to sham control in the retina (*p <* 0.05) (Figure 4B) and in the ONH region (Figure 4C). In the control retinal tissue, positive ZO-1 staining was apparent in the retinal INL and GCL. Only a few positive cells were found in the retina from irradiated mice (Figure 5A). Decreased ZO-1 expression was detected in the retina for both single and fractioned dose groups compared to controls (*p <* 0.05) (Figure 5B). However, positive staining for ZO-1 immunoreactivity, as observed in the retina of the control group, was not detected in the ONH region among all groups (data not shown).

## 4. Discussion

This study demonstrated that 0.5 Gy of proton radiation induced significant changes in oxidative stress, apoptosis, and BRB integrity biomarkers in the mouse retina four months after a dose of 0.5 Gy. There were differential responses between single and fractionated dose schedules in several parameters. Differences related to BRB biomarker distribution were observed between the retina and ONH (Table 1).

Radiation increases the activity of reactive oxygen species (ROS) [10]. Free radicals initiate a variety of cellular signaling and alter biomarkers that lead to cellular damage. 4-HNE is considered an oxidative stress biomarker for lipid peroxidation [11]. Studies suggest that HNE plays an important role in stress-mediated signaling for cellular damage and in the mechanisms of apoptosis caused by these stress factors [12]. Our observations show increased 4-HNE staining indicative of oxidative damage to the retina and optic nerve after proton radiation exposure compared to controls. The findings were consistent with previous findings in many research models [11,13]. Radiation delivered in fractionation over 25 days showed a smaller level of HNE expression than that of a single dose in the retina and ONH. However, single- and fractionated-dose schedules appear to produce a similar level of apoptosis in the retina, while fractionated dose schedules showed a higher number of apoptotic cell damage in the ONH region. It is the general consensus that for radiotherapy, and for patients who receive high total doses, a hyperfractionated dose produces less damage to normal tissue by permitting dose recovery for damage repair to DNA and other proteins [14]. However, our retina apoptosis data did not indicate that there was substantial repair of injury or dose recovery at a late observation time point of four months, although the total doses that were delivered to the mouse were much lower than the dose delivered to the patients in the radiotherapy. Our previous study also showed that there were no significant differences in retinal endothelial cell density following single or split dose with two equal dose fractions of 8–28 Gy [15]. Evaluation of DNA repair of double-strand breaks in the rabbit retina, confirms that little DNA repair for dose recovery was found in the aged rabbit retinas [16,17], although retinal radiosensitivity of mice may be different compared to that of rats or rabbits. Surprisingly, a higher number of TUNEL positive cells were detected in the fractionated dose group compared to that of acute single dose exposure in the ONH region. One possible explanation is that acute radiation-induced apoptosis in the ONH may precede fractionated doses-induced apoptosis. By four months, the level of apoptotic damage already passes its peak in the retina that received single dose of 0.5 Gy, so it is less pronounced in acute irradiated group than that of fractionated dose group at four months. However, the complex regional difference in response to single and hyperfractionated dose schedule warrant further investigation. The difference in structure, cellular composition, and permeability or BRB property may contribute to the capacity of the tissue to regulate radiation response in the retina and ONH.

In the present study, we found that low-dose proton radiation induced significant changes in the expression of several proteins related to BRB integrity, including APQ-4, PECAM-1, and ZO-1 in the retina. AQP-4, a water channel protein, has an important role in maintaining the integrity of BRB by forming a functional complex at astrocytic endfeet [18]. The upregulation of AQP-4 in irradiated retinal GCL and INL indicate the potential disruption of barrier integrity and even possible edema formation. PECAM-1 is constitutively expressed on endothelial cells and leukocytes that can be highly induced in response to inflammatory situations, leading to secretion of chemokines and cytokines [19]. PECAM-1 serves as an adhesion-signaling molecule with important roles in inflammation and leukocyte transmigration [20]. It functions as an adhesive stress-response protein to modulate endothelial cell junctional integrity [21] and may serve as a primary coordinator of many signaling pathways involved in cell metabolism and protein expression [22]. Increased expression of PECAM-1 has also been shown to lead to disruption of BRB integrity after retinal vascular injury [23]. Leukocyte migration from the blood into the retinal tissue promotes the breakdown of the BRB [24]. The higher PECAM-1 expression in the retina following radiation exposure observed in our study may facilitate inflammatory cells to cross the BRB due to changes in the BRB permeability [25]. The integrity of the BRB is also dependent on tight junctions (TJs). Tight junctions are transmembrane proteins and peripheral cytoplasmic proteins. ZO-1 is peripheral cytoplasmic protein and is critical in the initial formation, distinct organization, and function of TJs [26,27]. Many studies have shown that vascular permeability is altered by decreasing the levels of tight junction proteins, including ZO-1 [28]. Our previous spaceflight study also showed decreased immunoreactivity of ZO-1 expression in the retina of the flight group compared to the ground controls [29]. Targeted irradiation of mice caused corneal endothelial cell loss, reduction of ZO-1 junctional contacts, and induction of corneal edema [30]. The degradation of TJ-associated proteins in the retina following injury has been also shown to lead to BRB breakdown [18]. In the current study, the decrease in ZO-1 protein expression in the retina together with the increase in water channel protein AQP-4 and adhesion molecules PECAM-1, in conjunction with increased apoptosis, suggests the BRB disturbance and prolonged tissue remodeling due to radiation exposure with single or fractionated dose schedules. 

More recently, ONH has become one of the focuses in the research surrounding SANS [7]. Edema was observed in the prelaminar ONH in astronauts [31]. These data points to the hypothesis that multiple factors, including fluid shifts, may lead to ONH swelling and associated intraocular alterations [32,33]. Previous human and monkey studies showed that prelaminar ONH has distinct BRB properties when compared to the retina. Microvessels in the prelaminar region of the ONH lack blood-brain barrier (BBB) proteins and display nonspecific permeability, while the retina has BBB characteristic proteins but lacks permeability biomarkers [34]. Some of our results are in agreement with their findings. In our mouse study, we found that radiation exposure induced significant changes in the expression of several proteins related to BRB integrity and function, including APQ-4, PECAM-1, and ZO-1 in the retina. The staining pattern in the ONH differs from that of the retina. Low level positive staining for BRB biomarkers including water channel protein, AQP-4 and TJ protein, ZO-1 in ONH region indicates deficient BRB function, although adhesion molecule, PECAM-1 was present in the ONH. PECAM-1 also plays an important role in mediating trans-endothelial immune cell migration [35] and participates in key inflammatory events [25]. Lack of BRB biomarkers in the ONH may reflect dysregulation of barrier function and vessel integrity that can cause chronic inflammation, rendering more susceptible of this region to environmental insults and pathology of ONH diseases. Further studies are needed to investigate vascular permeability as well as lymphatic and glymphatic clearance related to the accumulation of fluid in the ONH region [36]. 

In conclusion, we hope that our investigation sheds light on the mechanism of space radiation-induced ocular response. Knowledge gained from this study could aid further approaches toward more effective countermeasures during human space travel. Our long-term goal is to use system biology approach to investigate the association between ophthalmic changes and other organs pathologies in response to ionizing radiation and to facilitate a better understanding of space radiation-induced tissues response and its relationship to health implication that may lead to disease development. 

## Figures and Tables

**Figure 1 life-11-00849-f001:**
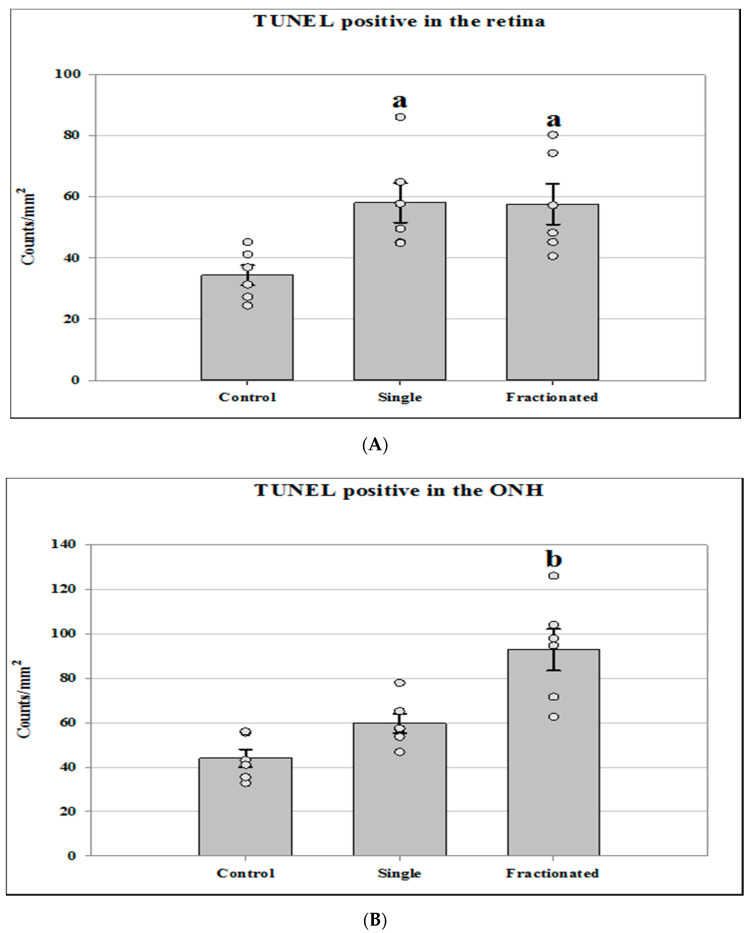
Apoptosis based on terminal deoxynucleotidyltransferase dUTP nick-end labeling (TUNEL) staining of male C57BL/6 following an acute single or fractionated (12 equal fractions) proton irradiation for a total dose of 0.5 Gy in mouse ocular tissue. (**A**) Apoptotic cell density in the retinal outer nuclear layer (ONL), inner nuclear layer (INL), and ganglion cell layer (GCL); (**B**) Apoptotic cell density in the optic nerve head (ONH) region. The density profiles were expressed as the mean number of apoptotic positive cells/mm^2^. The mean of the density profile measurements across 5 retina sections per eye was used as one experimental value. Values are represented as mean density ± SEM for 6 mice/group. ^a^ Significantly higher than controls (*p* < 0.05). ^b^ Significantly higher than all other groups (*p* < 0.05).

**Figure 2 life-11-00849-f002:**
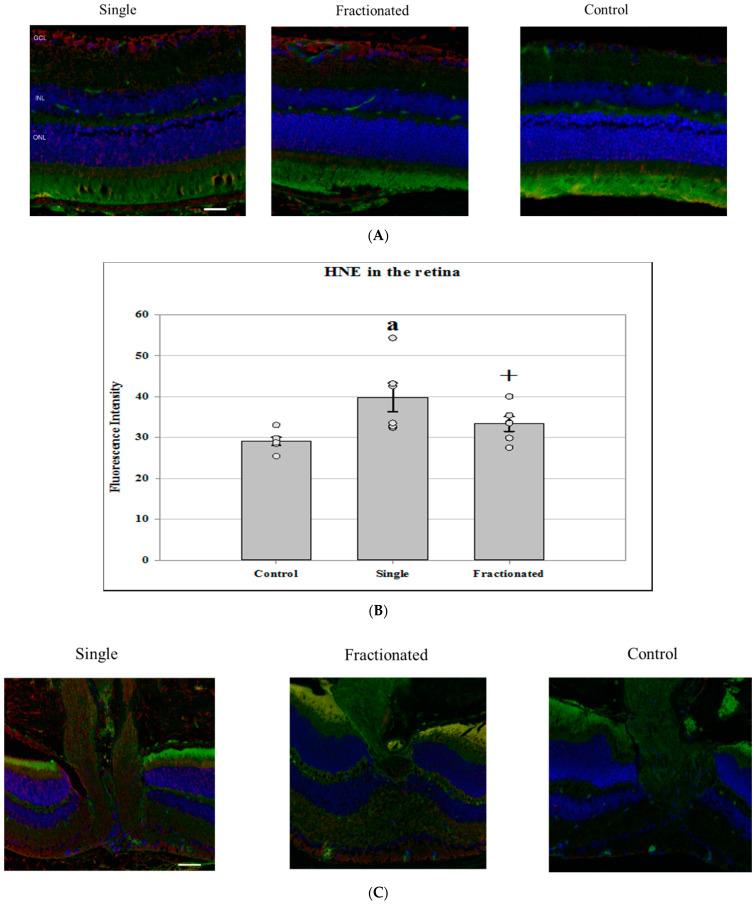
Cellular oxidative damage in the retina and optic nerve head (ONH). (**A**) Representative micrographs of ocular sections were evaluated for lipid peroxidation by immunostaining with anti-4-hydroxynonenal (4-HNE) antibody in the retina of irradiated and control samples. 4-HNE positive staining was identified with red fluorescence; the nuclei were counterstained with DAPI (blue). The vessels were stained with tomato lectin (green). Scale bar = 50 μm. (**B**) The average fluorescence intensity for HNE in the retina was measured and calculated using the ImageJ program. Fluorescence was averaged across 5 ocular sections per eye as one experimental value. Values are represented as mean density ± SEM for 6 mice/group. ^a^ Significantly increased 4-HNE staining compared to control group (*p* < 0.05). ^†^ Higher than control with a strong trend (*p =* 0.06). (**C**) Representative micrographs of ocular sections were evaluated for lipid peroxidation by immunostaining with anti-4-hydroxynonenal (4-HNE) antibody in the ONH. (**D**) The average fluorescence intensity for HNE in the ONH was measured and calculated using the ImageJ program. Fluorescence was averaged across 5 ocular sections per eye as one experimental value. Values are represented as mean fluorescence intensity ± SEM for 6 mice/group. ^a^ Significantly higher than all other groups (*p* < 0.05). ^b^ Significantly higher than the control group (*p* < 0.05).

**Figure 3 life-11-00849-f003:**
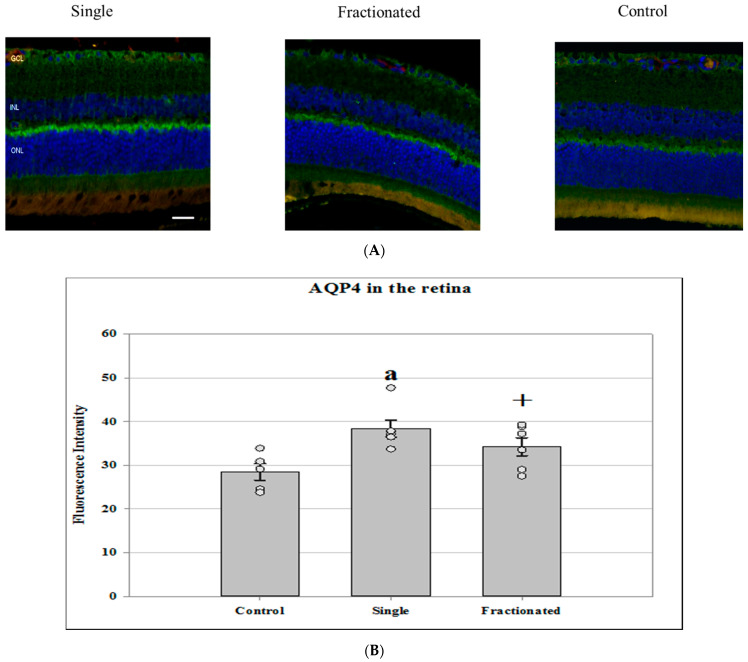
Glial fibrillary acidic protein (GFAP) and aquaporin-4 (AQP-4) staining in the retina and ONH. (**A**) Representative micrographs of ocular sections with anti-GFAP and AQP-4 antibodies in the retina of irradiated and control samples. AQP-4 positive staining is identified by green fluorescence, GFAP with red, and the cell nuclei with blue (DAPI). Scale bar = 50 μm. (**B**) The average fluorescence intensity for AQP-4 was measured in the retina and calculated using the ImageJ program. Fluorescence was averaged across 5 ocular sections per eye as one experimental value. Values are represented as mean fluorescence intensity ± SEM for 5–6 mice/group. ^a^ Significantly higher than control group (*p* < 0.05). ^†^ Higher than controls with a strong trend (*p =* 0.07).

**Figure 4 life-11-00849-f004:**
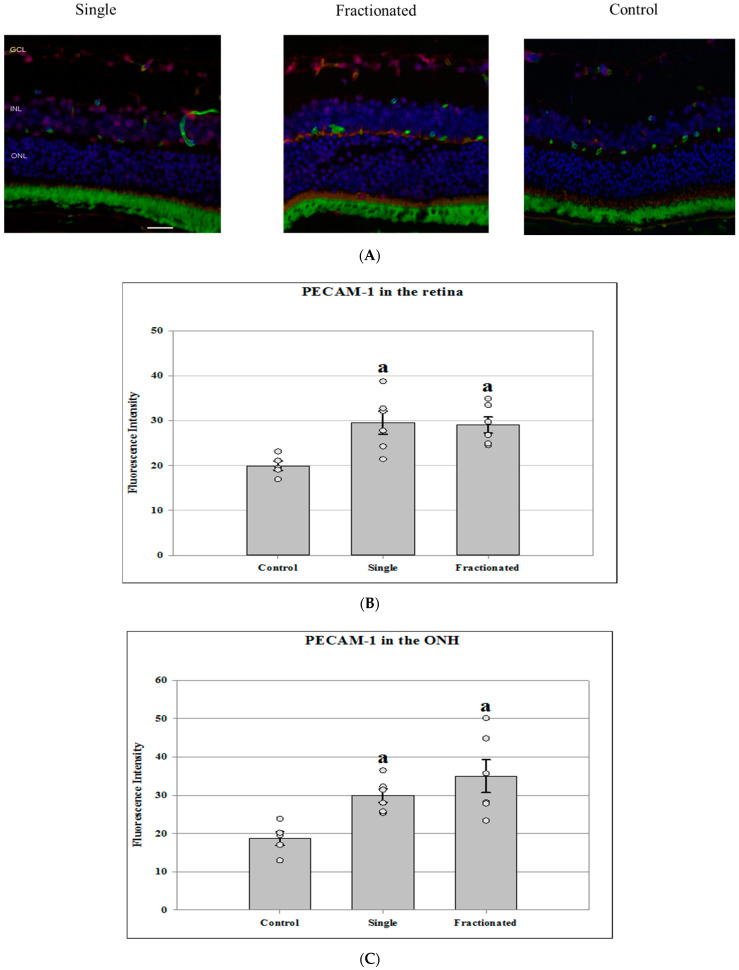
Platelet endothelial cell adhesion molecule (PECAM-1) staining in the retina and ONH. (**A**) Representative images of PECAM-1 ocular sections in the retina of proton irradiated and control mice. PECAM-1 positive cells were identified with red fluorescence, endothelium was stained with lectin (green). The nuclei of photoreceptors were counterstained with DAPI (blue). Scale bar = 50 μm. (**B**) Immunoreactivity of PECAM-1 staining in the retina. The average fluorescence intensity for PECAM-1 activity was measured and calculated using the ImageJ program. Fluorescence was averaged across 5 ocular sections per eye as one experimental value. Values are represented as mean fluorescence intensity ± SEM for 5–6 mice/group. ^a^ Significantly higher than controls in the retina (*p* < 0.05). (**C**) Representative images of PECAM-1 ocular sections in the retina of proton irradiated and control mice. ^a^ Significantly higher than control group in the ONH (*p <* 0.05).

**Figure 5 life-11-00849-f005:**
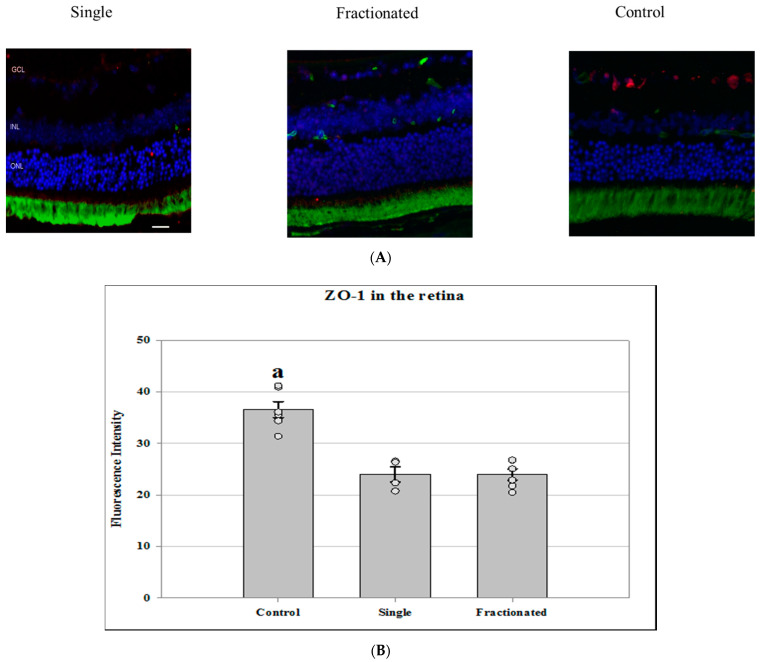
Zonula occludens-1 (ZO-1) staining in the retina and ONH. (**A**) Representative images of ZO-1 in ocular sections of proton irradiated and control mice. ZO-1 positive cells were identified with red fluorescence, endothelium was stained with lectin (green). The nuclei of photoreceptors were counterstained with DAPI (blue). Scale bar = 50 μm. (**B**) Immunoreactivity of ZO-1 staining in the retina. The average fluorescence intensity for ZO-1 was measured and calculated using the ImageJ program. Fluorescence was averaged across five retinas per group as one experimental value. Values are represented as mean fluorescence intensity ± SEM for 5–6 mice/group. ^a^ Significantly higher than other groups (*p <* 0.05).

**Table 1 life-11-00849-t001:** Summary of Biomarker Expressions in The Retina and Optic Nerve Head (ONH).

Biomarkers	Retina	ONH
TUNEL	Single ↑ ≈ fractionated ↑	Single ↔ < fractionated ↑
4-HNE	Single ↑ > fractionated ↑	Single ↑ > fractioned ↑
AQP-4	Single ↑ > fractioned ↑	Single ↔ ≈ fractioned ↔
PECAM-1	Single ↑ ≈ fractionated ↑	Single ↑ ≈ fractionated ↑
ZO-1	Single ↓ ≈ fractionated ↓	Single ↔ ≈ fractioned ↔

TUNEL: Terminal deoxynucleotidyl transferase dUTP nick end labeling; 4-HNE: 4-hydroxynonenal; AQP-4: aquaporin-4; PECAM-1: platelet endothelial cell adhesion molecule; ZO-1: Zonula occludens-1. ↑ Significant (*p* < 0.05) or trend increase compared to controls. ↓ Significant (*p <* 0.05) decrease compared to controls. ↔ No changes compared to controls.

## Data Availability

The data presented in this study are available on request from the corresponding author.

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
