# Peer review of "Evaluating Ocular Response in the Retina and Optic Nerve Head after Single and Fractionated High-Energy Protons"

_life, 2021, doi:10.3390/life11080849_

Round 1

Reviewer 1 Report

This manuscript by Mao et al. studied the effects of acute vs. fractionated high energy proton radiation on mice retina and optic nerve head. The authors have shown differential molecular biomarker signatures of acute and fractionated proton exposures. Results presented in this manuscript are interesting and are important to understand the role of space radiation exposure in neuro-ophthalmic abnormalities during and after deep space exploration, however, the extent of the study end-point is limited to a single time point.  The authors should address the following concerns during the revision of this paper: 

  1. Provide make/model of the proton source. 
  2. Provide LET (or a range) in the M&M for the spread out bragg peak and proton energy (0-140 MeV/n) used in this study. 
  3. "Low-dose" is a relative term and has misleading interpretations in radiobiology, therefore needs to be deleted throughout the manuscript. 
  4. Add "space" between radiation dose and unit ["0.5Gy" should be written as "0.5 Gy"]
  5. The authors should re-plot all the graphs to keep control bars on the left side followed by treatment groups. 

Reviewer 2 Report

Figure 1A: edit "postive" into "positive"

Author Response

Figure 1A: edit "postive" into "positive"

Response: Thank reviewer for pointing that out. The word is corrected.

Reviewer 3 Report

Comments/Questions :

  • Line 72 : Can you explain the use of 6 months old mice ? Often 8 to 12 weeks old mice are used for total body irradiation experiment.
  • Line 79 : Can you give us the composition of the spread out bragg peak and the positions of the boxes in spread out bragg peak.
  • Line 81 :Can you explain precisely how the dosimetry was made ? Use of animals restrains CT scan with TPS, MC calculations, measure ? Can you precise the modality of protonbeam delivery (pencil beam scanning, double scaterring) and the provider of the beam source ?
  • Line 93 : It is unusual to use freely-moving animals restrains cage for proton irradiation as you will increase interface air/matter in the beam and the position of the animal will change in the beam during the treatment that will increase dose distribution heterogeneity. Can you explain this choice and precise the uncertainty of dose received linked to this set-up ? Can you provide a picture of the set-up ?
  • Line 95 : can you precise the size of the beam ?
  • Line 111 : Explain why do you don’t use flat-mount of retina that his better to analyse BRB breakdown such as with FITC-Dextran ? (https://www.researchgate.net/publication/321258394_Aster_koraiensis_extract_prevents_diabetes-induced_retinal_vascular_dysfunction_in_spontaneously_diabetic_Torii_rats/figures?lo=1)
  • Line 155 : It is better to normalize the number of tunel positive cells to the nomber of cells int the tissue as i twill take into account variation of cells in tissue. Please can you provide this data ? If it is easier for you, youn can provide the density of cells per mm² and normalize the number tunel positive cells per mm² with it. I will expect to observe differences in cell density after irradiation.
  • Line 158 : ImageJ 1.4V is an old version of the software (before 2011). Is that correct or an error of copy and paste ?
  • Line 207 : Can you improve picture resolution ?
  • Line 219 : Do you already use unstained tissue to assess autofluorescence and determine intensity fluorescence values ? How do you determine precisely fluorescence intensity ? Do you use tissues area to normalize it ? If it is not Can you normalize it ?
  • Line 219 : If i split the channel and do a quick analysis with ImageJ, i find more red intensity in control than for irradiated sampe. Do you use an auto display for this image ? Can you explain precisely how the analysis was made and show splitted channel images ? Can you use the same display for all images and channels ? It will be the same question for all the figures.
  • Line 233 : Can you present the quantification for GFAP ?
  • Line 296 : Lack of a word (… study).
  • Line 325 : Previous experiments was done with rats using high doses. As the tissues radiosensitivity of rats are different compared to mice tissues radiosensitivity, you cannot do such extrapolation. Further more in this study the dose use dis in the range that induce Low-dose hypersensitivity. Then you have to demonstrate that it is true in your conditions.
  • Line 362 : ref 29 is not relevant as they use UVA instead of ionizing radiations.

  • Do you already assess BRB integrity using fonctionnal assays such as blue evans, fluorescein angiography, albumin leakage, FITC-Dextran ? If it is not the case explain why.
  • Can you explain the choice of the different assays and the link between chosen markers modulation and neuro-ophthalmic abnormalities expected ?
  • Can you evaluate the number amount of pericytes as they are involved in the BRB integrity ?
  • Figures 1,2,3,4 and 5 :Each animals should be presented as single dots with the median in order to assess variability.
  • Present and discuss the Low-dose hypersensitivity phenoma.
  • As other organ will become pathological in your experiment such as heart (cardiac hypertrophy). Can you discuss the possibility to have a link between ophtalmic abnormalities and other organs pathologies. In fact, in the case thoracic irradiation, if the heart is irradiated, lung injuries will be improved.

It is an original study as very few people are studying the effects of ionizing radition on retina. Further more, experiments like this are very difficult to do according to the accessibility of beam and the problems linked to the use of medical device for animal experiments.
